# TOWARD EDITABLE VECTOR GRAPHICS: LAYERED SVG SYNTHESIS FROM MULTIMODAL PROMPTS

## ABSTRACT

Scalable Vector Graphics (SVGs) are essential for modern design workflows, yet existing methods are confined to single-modality inputs and produce non-editable outputs. To bridge this gap, we introduce LayerVec, the *first* framework to synthesize editable, layered SVGs from multimodal prompts. LayerVec is designed to operate on top of powerful Unified Multimodal Models (UMMs), employing a dual-stage pipeline: it first generates a raster guidance image, then uses an iterative deconstruction process to intelligently segment it into semantically coherent vector layers. To facilitate rigorous evaluation, we conduct MUV-Bench, a comprehensive benchmark, and Layer-wise CLIP Consistency (LCC), a metric assessing structural editability. Experiments show LayerVec significantly outperforms state-of-the-art baselines in producing structurally clean and semantically accurate SVGs. We further demonstrate its robustness and model-agnostic nature by showing consistent performance gains across different UMM backbones.

## 1 INTRODUCTION

Scalable Vector Graphics (SVG) are vital for modern image rendering due to their resolution independence and editability (Polaczek et al., 2025; Yang et al., 2025; Song & Zhang, 2022). This structure enables precise, non-destructive editing at the object level, where layered organization is crucial for managing complex scenes. In professional design pipelines, different elements are conventionally separated into <g> groups (as visualized in Fig. 1), allowing designers to manipulate objects independently and maintain semantic clarity (Zhang et al., 2025). However, manually crafting SVGs with such structured layers remains a significant challenge, demanding extensive time and technical expertise (Jain et al., 2023; Wang et al., 2025). This challenge drives the urgent need for automated methods that can translate high-level creative intent into production-quality, layered, and editable vector graphics.

Existing methods adopt either optimization-based methods (Xing et al., 2024; Jain et al., 2023; Song et al., 2025; Mirowski et al., 2022; Frans et al., 2022) or LLM-based approaches (Yang et al., 2025; Wu et al., 2024) to generate SVG contents. Despite their effectiveness, as shown in Fig. 1 all existing SVG generation methods encounter a ***fundamental limitation***: *they operate exclusively on single-modality inputs (text-only or image-only)*, creating a striking disconnect from real-world design workflows that inherently rely on multi-modal inputs combining visual references with textual instructions (Ye et al., 2025; Kawar et al., 2023). A reference raster image combined with an editing instruction provides a more accurate and intuitive specification than text alone (Fu et al., 2024; Bai et al., 2025a), since visual examples are both common and expressive than text alone. However, this critical capability remains entirely unexplored in current SVG generation literature.

To bridge this gap, we introduce LayerVec, the **first** framework to synthesize SVG from multimodal inputs. Our approach employs a two-stage pipeline. In the first stage, we generate a guidance raster image that fuses visual references with textual instructions, creating a unified representation that captures both visual context and editing intent(See Sec. 3.2). In the second stage, we propose an iterative method that progressively extracts semantic entities from the guidance image and converts them into distinct vector layers through post-vectorization(See Sec. 3.3). This process transforms the raster guidance into a structured SVG file while ensuring coherent scene decomposition, preserving object-level semantics, and enabling fine-grained layer-wise editing. The framework produces

Figure 1: **Visualization Results of Our Proposed LayerVec**. Our framework enables layered and editable SVG generation from multimodal inputs, addressing the limitations of prior single-modality methods. **Left:** LayerVec handles multimodal inputs into semantically layered SVG file. **Right:** Comparison against traditional single-modality approaches (text-only or image-only), where our method (b) produces clean and well-ordered layers, while prior methods (a) yield flat or entangled outputs.

multi-layered vector graphics that uniquely combine high visual fidelity with semantic consistency, establishing a new paradigm for multimodal vector graphics generation.

Moreover, to enable rigorous evaluation of this unexplored task, we proposing MUV-Bench (Multimodal-to-Vector Benchmark), the *first comprehensive benchmark* specifically designed for multimodal-to-SVG generation. MUV-Bench comprises 50 diverse raster images paired with 10 professional editing instructions each, resulting in 500 carefully curated tasks spanning five essential categories: Object Addition, Object Removal, Motion Change, Background Change, and Style Transfer.

Furthermore, we identify a *fundamental misalignment* in existing evaluation metrics of structured vector generation. Furthermore, we identify a fundamental flaw in existing metrics: they evaluate the final rendered image but ignore the crucial layered structure that defines editable vector graphics. This allows models with visually plausible but structurally entangled outputs to score highly. To address this, we propose Layer-wise CLIP Consistency (LCC), a novel metric that directly assesses editability by computing the CLIP similarity (Frans et al., 2021) between each semantic layer and its corresponding label(See Sec. 4.1). Comprehensive experiments have demonstrated that our LayerVec significantly outperforms both open- and closed-source baselines across text-to-SVG and multimodal-to-SVG tasks. Additionally, we validate the strong generalization capability of the proposed framework across different backbone architectures(See Sec. 4.5).

**In summary, our contributions are as follows:**

1. To the best of our knowledge, this paper is the *first* to explore SVG generation from multimodal inputs. We introduce LayerVec, a novel two-stage framework that first synthesizes a high-fidelity raster image from multimodal inputs, then decomposes it into semantically coherent layers. This enables the generation of semantically structured, multi-layered SVGs, bridging a critical gap in vector graphics generation.

2. We introduce MUV-Bench (Multimodal-to-Vector Benchmark), the first comprehensive evaluation benchmark comprising 500 diverse multimodal tasks, enabling systematic and standardized assessment of multimodal SVG generation methods.

3. We identify a fundamental misalignment in existing evaluation metrics that ignore the structural aspects crucial to vector graphics quality. To address this, we propose Layer-wise CLIP Consistency (LCC), a novel metric that computes CLIP similarity between each raster layer and its corresponding semantic label, serving as an automated proxy for structural editability.

4. Comprehensive experiments have demonstrated that our framework achieves significant improvements over strong baselines, successfully handling multimodal inputs to generate both simple icons and complex illustrations with high fidelity. Moreover, our approach exhibits robust generalization, transferring effectively across different Unified Multimodal Model backbones, establishing its broad applicability and reliability.

## 2 RELATED WORK

### 2.1 TEXT-GUIDED SVG GENERATION

Early approaches use the Sequence-To-Sequence (seq2seq) architecture to generate SVGs (Carlier et al., 2020; Ha & Eck, 2018; Reddy et al., 2021; Wang & Lian, 2021; Wang et al., 2022). Nonetheless, the absence of large-scale vector datasets constrains their generalization capabilities and the creation of complex graphics, with most datasets focusing on specific areas like monochromatic vector icons (Wu et al., 2023a) and fonts (Wang & Lian, 2021; Song & Zhang, 2022). Optimization-based methods (Thamizharasan et al., 2024; Xing et al., 2024; Jain et al., 2023; Song et al., 2025; Mirowski et al., 2022; Frans et al., 2022) directly optimize SVG paths by leveraging powerful priors from vision-language models such as CLIP or diffusion models (Rombach et al., 2022), yet are critically hampered by prohibitive computational costs and the production of structurally fragmented, poorly editable vector outputs. More recently, the advent of Large Language Models (LLMs) unleashes the potential of generating SVGs via XML code generation (Wu et al., 2024; Yang et al., 2025; Xing et al., 2025). These methods, however, are often constrained by context length and require post-processing to approximate a layered structure, undermining the goal of structured generation.

Crucially, professional vector graphics require multi-layer structures so that individual objects can be independently edited, reused, or recomposed without entanglement. Existing methods, however, either collapse into fragile shapes or produce incoherent groupings, which severely limits practical usability. To address this, we propose a novel raster-to-vector deconstruction pipeline that is natively multimodal: guided by an MLLM's semantic hierarchy, our framework translates complex multimodal prompts into clean, layered SVGs that align with real-world design workflows.

### 2.2 UNIFIED MULTIMODAL MODELS

Unified multimodal models has attracted significant attention in recent years (Wu et al., 2025b; Bai et al., 2025b; Mao et al., 2025). For instance, OmniGen (Xiao et al., 2025) employs a streamlined Transformer architecture to address diverse image generation tasks without requiring additional plugins or preprocessors. Recent breakthroughs, such as Gemini-2.0-flash (Google, 2025) and GPT-4o (OpenAI et al., 2024), have underscored the field's vast potential, signaling a paradigm shift from specialized models (Bai et al., 2025b; Black-Forest-Labs, 2024) towards powerful, unified multimodal systems. While models like Chameleon (Chameleon, 2025) and Emu3 (Wang et al., 2024) utilize discrete autoregressive methods across modalities, the Janus series (Wu et al., 2025a) employs dual image encoders for both understanding and generation tasks.

## 3 METHODOLOGY

In this section, we present LayerVec, our unified framework for semantic layer-wise SVG generation. As illustrated in Fig. 2, our pipeline consists of two parts: (a) Raster Guidance generation and (b) Layered SVG generation.

### 3.1 PRELIMINARIES

**Problem formulation.** Our objective is to map a multimodal prompt $P = \{P_T, I_R\}$, consisting of a textual description $P_T$ and an optional reference image $I_R$, to a layered Scalable Vector Graphic (SVG). The target SVG is a structured document, defined as an ordered set of layers $L = (l_1, \ldots, l_N)$ representing the scene's Z-order. Each layer $l_i$ corresponds to a distinct semantic entity and is composed of geometric primitives. The core task is to learn a mapping $\mathcal{F}$, such that:

$$L = \mathcal{F}(P) \quad \text{s.t.} \quad \mathcal{R}\left(\bigcup_{i=1}^{N} \theta_i\right) \approx \mathcal{I}(P), \tag{1}$$

Figure 2: **The LAYERVEC framework architecture.** The first stage, Raster Guidance Synthesis, generates a high-fidelity image $I_0$ from multimodal inputs $P = \{P_T, I_R\}$. The second stage, Layered SVG Synthesis, iteratively deconstructs $I_0$ into semantic raster layers which are then vectorized into the final layered SVG.

where $\mathcal{R}(\cdot)$ is a differentiable renderer converting path parameters $\theta_i$ into a raster image, and $\mathcal{I}(P)$ is the ideal visual interpretation of the prompt.

**Foundational model.** Our framework is designed to be model-agnostic, leveraging the capabilities of any powerful Unified Multimodal Model (UMM) that provides both visual understanding and image synthesis functionalities. For our primary experiments, we build upon OmniGen2 (Wu et al., 2025b), which features a decoupled architecture with a distinct MLLM for reasoning ($\mathcal{M}$) and a Diffusion Transformer (DiT) for generation ($\mathcal{D}_\theta$). To adapt the model for the unique vector-graphics aesthetic, we specialize its synthesis component. We freeze the core understanding modules (like OmniGen2's MLLM) and apply lightweight LoRA (Hu et al., 2021) fine-tuning only to the image generation engine (i.e., the DiT) using a flow-matching objective on a curated vector-style dataset:

$$\mathcal{L}_{\mathrm{FM}}(\theta + \Delta\theta) = \mathbb{E}_{t,x_0,x_1} \|\mathbf{v}_{\theta+\Delta\theta}((1-t)x_0 + tx_1) - (x_1 - x_0)\|_2^2 , \qquad (2)$$

Details of the training procedure are provided in App. F.1. This process yields a specialized generator adept at creating high-quality raster priors. To validate our framework's model-agnostic claim, we demonstrate its effectiveness on another distinct UMM architecture, BAGEL-7B, in Sec. 4.5.

### 3.2 RASTER GUIDANCE GENERATION

The initial step in our pipeline is the generation of a high-fidelity raster image, $I_0$, which serves as the visual anchor for the subsequent deconstruction. This stage leverages our foundational model to address two key challenges: resolving instructional ambiguity and preserving coherence during editing. To do so, we adapt established techniques for instruction decomposition and guided generation, integrating them directly into our MLLM-DiT architecture.

At inference, the MLLM $\mathcal{M}$ first interprets the multimodal prompt $P$ to produce contextual hidden states $\mathbf{h}$. These states, encapsulating the high-level semantic intent, are then fed into our fine-tuned Diffusion Transformer (DiT), $\mathcal{D}_{\theta'}$, to guide the generation process. This yields a high-fidelity raster image $I_0$ that serves as a visual prior for the subsequent deconstruction.

### 3.3 LAYERED SVG GENERATION VIA SCENE DECONSTRUCTION

**Iterative Scene Deconstruction.** Given the raster prior $I_0$ synthesized in Stage 1, we deconstruct it into a layered representation through an *Iterative Scene Deconstruction* process. This process is orchestrated by MLLM $\mathcal{M}$, which functions as a high-level scene planner. Conditioned on the instruction $P_T$ and the image $I_0$, it employs chain-of-thought reasoning to identify a sequence of foreground entities $E = \{e_1, \ldots, e_{N-1}\}$ and determines their front-to-back Z-order, $O = \{o_1, \ldots, o_{N-1}\}$.

The framework then iteratively processes these entities, beginning with the initial image $I_0$. As shown in Fig. 3, in each iteration $i$, the MLLM directs the extraction of entity $o_i$ from the current image $I_{i-1}$ in two steps. First, it proposes a bounding box $B_i$ to guide a segmentation model $\mathcal{S}$, yielding a mask $m_i = \mathcal{S}(B_i, I_{i-1})$. This mask is used to extract the corresponding raster layer, $\hat{l}_i = I_{i-1} \odot m_i$. Subsequently, to address the occlusion created by this extraction, the MLLM generates a context-aware inpainting prompt $P_{\mathrm{inp}}^{(i)}$. A dedicated diffusion inpainting model $\mathcal{D}_{\mathrm{inp}}$ then synthesizes the missing region, producing the next image state $I_i = \mathcal{D}_{\mathrm{inp}}(I_{i-1}, m_i, P_{\mathrm{inp}}^{(i)})$.

This process repeats until all $N-1$ entities are extracted. The final image state, $I_{N-1}$, becomes the background layer $\hat{l}_{\text{bg}}$. The result is a complete set of raster layers, $\hat{\mathcal{L}} = \{\hat{l}_1, \ldots, \hat{l}_{N-1}, \hat{l}_{\text{bg}}\}$, ready for the final vectorization stage. In summary, this process's detailed procedure is shown in App. E.

**Vectorization and composition.** In the final stage, we convert the deconstructed raster layers $\hat{\mathcal{L}}$ into a structured SVG. For this, we employ VTracer (Sanford Pun, 2020), as the image vectorization engine, to process each raster layer $\hat{l}_i$ independently. At a high level, VTracer operates by first tracing the boundaries of color regions in the raster layer to form initial polygons. It then intelligently simplifies these polygons by removing redundant vertices and fits the simplified boundaries with smooth Bézier curves. This process yields compact and faithful vector representations $l_i$ for each layer. Crucially, by applying this powerful vectorizer to our semantically pre-separated raster layers, we ensure that the final SVG maintains the clean, editable, and meaningful structure established during our iterative deconstruction.

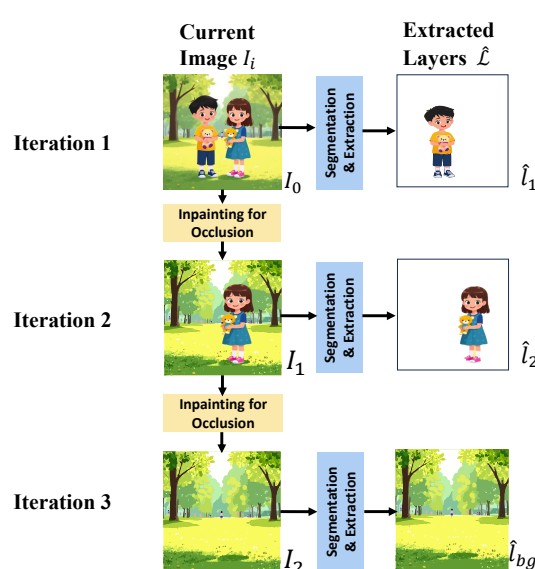

Figure 3: **The process of our Iterative Scene Deconstruction.** Our pipeline iteratively "peels" the raster image ($I_0$) from front to back. In each step, it segments and extracts the foremost object into a layer ($\hat{l}_i$) and then inpaints the remaining image ($I_i$) to handle occlusions.

Finally, the complete SVG document $L$ is assembled by composing all generated vector layers. The layers are stacked in reverse order of extraction, thereby preserving the Z-order hierarchy inferred during the deconstruction stage. This hierarchical composition is represented as:

$$L = \text{Merge}(l_{\text{bg}}, l_{N-1}, \ldots, l_1). \tag{3}$$

The outcome is a fully structured and editable vector graphic, enabling object-level manipulations such as translation, scaling, and recoloring in standard vector graphics software.

## 4 EXPERIMENTS

To comprehensively validate the effectiveness of LayerVec, we first describe the experimental setup in Sec. 4.1. We then present both quantitative and qualitative comparisons with state-of-the-art methods in Sec. 4.2 and Sec. 4.3, respectively. Furthermore, we conduct a user study in Sec. 4.4 to assess the editability of the generated SVGs. Finally, we provide evaluation studies in Sec. 4.5 to analyze the contribution of different components within our framework.

### 4.1 EXPERIMENTAL SETUP

**Baselines.** We compare our proposed method with various baseline approaches.

- *Text-to-SVG baselines:* We compare LayerVec with state-of-the-art approaches spanning both optimization-based and auto-regressive methods. For optimization-based methods, we evaluate against NeuralSVG (Polaczek et al., 2025), VectorFusion (Jain et al., 2023), SVGDreamer (Xing et al., 2024), and LayerTracer (Song et al., 2025). For LLM-based approaches, we compare with OmniSVG (Yang et al., 2025). Additionally, we assess a range of multimodal large language models (MLLMs) including open-source models (Deepseek-V3.1 (DeepSeek-AI, 2025) and Qwen2.5-VL-32b-Instruct (Bai et al., 2025b)) and closed-source models (Gemini 2.5 Pro (Google, 2025) and GPT-4o (OpenAI et al., 2024)), representing the most advanced closed-source multimodal capabilities.
- *Multimodal-to-SVG baselines:* Given the absence of methods specifically designed for multimodal-to-SVG generation, we evaluate LayerVec against the same MLLMs mentioned above, prompting them to generate layered SVGs with `<g>` structures from multimodal inputs.

**Evaluation Benchmarks.** For the Text-to-SVG task, we follow previous works (Wang et al., 2025; Yang et al., 2025) and evaluate on the MMSVG-Bench, which includes two subsets: *MMSVG-Icon* for simple objects and *MMSVG-Illustration* for complex scenes. As no standard benchmark exists for the multimodal-to-SVG task, we introduce our own: **MUV-Bench** (Multimodal-to-Vector Benchmark). It comprises 500 carefully curated tasks created from 50 source images and 10 editing instructions each. The benchmark covers five key professional editing categories: *Object Addition, Object Removal, Motion Change, Background Change,* and *Style Transfer*.

**Metrics.** For text-to-SVG task, following prior works (Xing et al., 2024; Jain et al., 2023; Yang et al., 2025), we use CLIP Score to measure how well the rendered SVG aligns with the input text, and use LAION aesthetic predictor (Wang et al., 2022) to estimate the aesthetic appeal of outputs. Moreover, we use HPS (Wu et al., 2023b) to evaluate our approach from a human aesthetic perspective.

Besides all the above metrics operating only on the raster image, we introduce Layer-wise CLIP Consistency (LCC). Given $K$ layers with semantic labels $\{e_1, \ldots, e_K\}$ and corresponding rasterizations $\{\hat{l}_1, \ldots, \hat{l}_K\}$, LCC computes the average CLIP similarity:

$$\text{LCC} = \frac{1}{K} \sum_{i=1}^{K} \text{CLIP}(\hat{l}_i, e_i). \tag{4}$$

A high LCC indicates that each layer faithfully captures its intended semantic entity, ensuring that vector paths grouped within SVG `<g>` blocks are both visually coherent and semantically interpretable. This makes LCC a direct and automatable proxy for the editability and structural soundness of the final SVG, complementing conventional image-level metrics.

For the multimodal task, we use three metrics. Following the protocol of GEdit-Bench (Liu et al., 2025), we employ GPT-4o (OpenAI et al., 2024) as an automated judge to score the rendered outputs on a 0–10 scale for: *(1) Semantic Fidelity (SF)*, measuring instruction alignment, and *(2) Structural Coherence (SC)*, assessing the visual similarity between non-edited regions of the output and the original image. Additionally, we use our proposed *Layer-wise CLIP Consistency (LCC)* to evaluate the semantic purity and editability of the generated layers.

## 4.2 QUALITATIVE COMPARISONS

Fig. 4 and Fig. 6 provide qualitative comparisons. On text-to-SVG tasks, we observe that LAYERVEC consistently produces visually faithful and structurally coherent SVGs across both *Icon* and *Illustration* domains. In contrast, optimization-based methods (e.g., SVGDreamer, VectorFusion) frequently yield noisy, over-complex curves with entangled paths that compromise editability, while LLM-based approaches (e.g., GPT-4o, Gemini2.5-Pro) often degenerate into oversimplified geometric shapes, particularly under complex illustration prompts.

Fig. 5 further emphasizes this point: compared to the disorganized curves from SVGDreamer and the overly primitive decomposition from Gemini2.5-Pro, LAYERVEC outputs well-layered SVGs with clean object boundaries and visually appealing design, making them both semantically accurate and directly usable in real editing workflows.

On multimodal-to-SVG, LayerVec effectively handles diverse editing operations, including motion edits, object addition, background replacement, and removal, while preserving scene coherence and editability. In contrast, LLM-based baselines struggle to generate code for complex illustrations, leading to incomplete or oversimplified edits. Optimization-based methods are not applicable in this setting due to their reliance on text-only inputs.

## 4.3 QUANTITATIVE COMPARISONS

Tab. 1 presents results on the text-to-SVG task. Existing LLM-based methods (e.g., GPT-4o, Gemini2.5-Pro) reach competitive CLIP scores on simple icons, showing basic text-image alignment, but their outputs degrade on complex illustrations, where oversimplified SVG code leads to poor Aesthetic and HPS scores. Optimization-based approaches (e.g., VectorFusion) generate more detailed images but produce tangled paths without meaningful layer separation, making them incompatible with LCC evaluation. By contrast, LayerVec achieves strong CLIP alignment across both icons

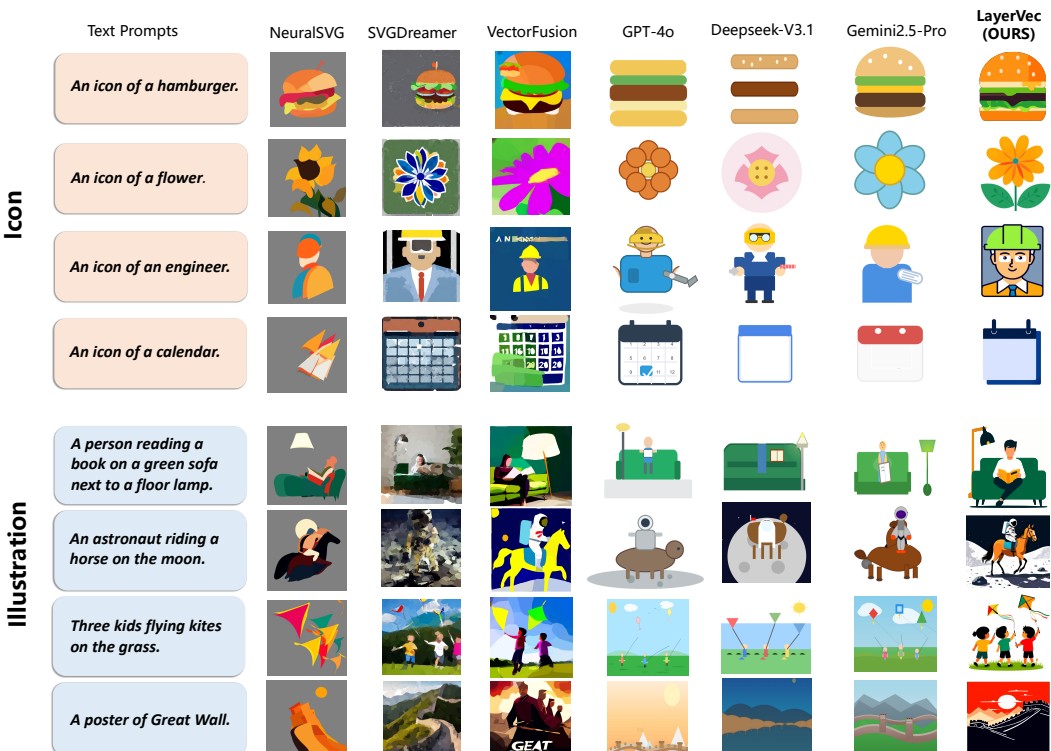

Figure 4: **Qualitative comparison with SOTA methods on Text-to-SVG task.** We compare the propose method with SOTA Text-to-SVG methods on both icons and illustrations. For simple Icons, baseline methods often produce either overly abstract shapes (OmniSVG, GPT-4o) or noisy results with redundant paths (SVGDreamer, VectorFusion). Our LayerVec consistently generates semantically accurate and structurally clear vector graphics across both domains.

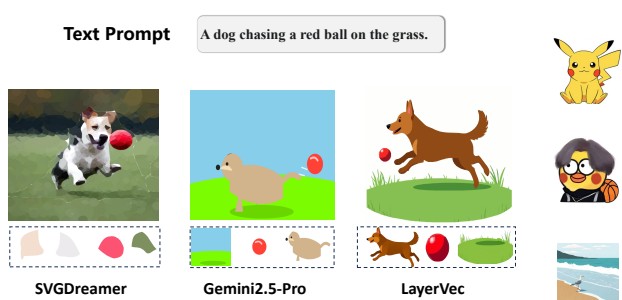

Figure 5: **Comparision on structure of output.** Optimization-based methods like SVGDreamer achieve visual complexity but decompose into a redundant "path soup". LLM-based methods like Gemini achieve object separation at the cost of extreme simplification. LayerVec is unique in its ability to produce an output that is both aesthetically rich and structurally sound.

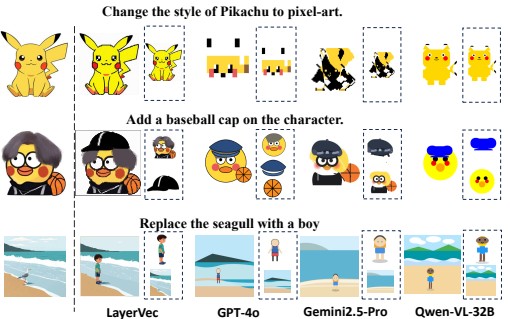

Figure 6: **Comparison of LayerVec and other methods on multimodal-to-SVG tasks.** Our method demonstrates superior performance in both instruction alignment and visual similarity, while uniquely providing a deconstructed, layered output for subsequent editing.

and illustrations, while also attaining the highest Aesthetic and HPS scores, indicating both visual quality and human preference. Crucially, it is the only method to obtain consistently high LCC values, demonstrating that its decomposed layers are semantically pure and structurally coherent—direct evidence that the generated SVGs are not only visually appealing but also truly editable.

Tab. 2 further evaluates the multimodal-to-SVG task across five editing categories. LLM-based baselines struggle to generate code for complex illustrations, leading to incomplete or oversimplified edits and low SF/SC scores. By contrast, LayerVec consistently outperforms all baselines, confirming its effectiveness for multimodal-to-SVG synthesis.

Table 1: Quantitative comparison for Text-to-SVG tasks. **Bold** and underline indicate the best and second-best results, respectively. Optimization-based methods cannot be evaluated on LCC due to their entangled paths without meaningful layer separation.

| Evaluation Dataset | Methods | Model Type | CLIP | Aesthetic | HPS | LCC |
|---|---|---|---|---|---|---|
| MMSVG-Icon | LayerTracer | Optim-Based | 0.2970 | 3.8101 | 0.1776 | N/A |
| | Vectorfusion | Optim-Based | 0.3041 | 4.1832 | 0.1885 | N/A |
| | SVGDreamer | Optim-Based | 0.2926 | 4.2740 | 0.2004 | N/A |
| | NeuralSVG | Optim-Based | 0.3008 | 3.9062 | 0.1982 | N/A |
| | Gemini2.5-Pro | LLM-Based | **0.3312** | 4.3207 | 0.1993 | 0.2302 |
| | Deepseek-V3.1 | LLM-Based | 0.3213 | 3.9153 | 0.1948 | 0.2568 |
| | Qwen2.5-VL-32b-Instruct | LLM-Based | 0.2732 | 3.5886 | 0.1914 | 0.2200 |
| | GPT-4o | LLM-Based | 0.3021 | 3.5684 | 0.1848 | 0.2543 |
| | OmniSVG | LLM-Based | 0.2194 | 3.1293 | 0.1739 | N/A |
| | LayerVec | this work | 0.3218 | **4.3836** | **0.2018** | **0.3035** |
| MMSVG-Illustration | LayerTracer | Optim-Based | 0.2748 | 3.6502 | 0.1776 | N/A |
| | Vectorfusion | Optim-Based | 0.3082 | 4.1079 | 0.1817 | N/A |
| | SVGDreamer | Optim-Based | 0.3407 | 4.4293 | 0.2060 | N/A |
| | NeuralSVG | Optim-Based | 0.3411 | 3.5886 | 0.1915 | N/A |
| | Gemini2.5-Pro | LLM-Based | 0.3329 | 4.0372 | 0.2119 | 0.2546 |
| | Deepseek-V3.1 | LLM-Based | 0.3188 | 3.5886 | 0.1946 | 0.2414 |
| | Qwen2.5-VL-32b-Instruct | LLM-Based | 0.2877 | 3.7664 | 0.1819 | 0.2301 |
| | GPT-4o | LLM-Based | 0.3076 | 3.5625 | 0.1883 | 0.2419 |
| | OmniSVG | LLM-Based | 0.2029 | 3.0156 | 0.1643 | N/A |
| | LayerVec | this work | **0.3445** | **4.5872** | **0.2187** | **0.3169** |

Table 2: **Multimodal-to-SVG results on MUV-Bench.** LayerVec excels in both Semantic Fidelity (SF) and Structural Coherence (SC).

| Task | Models | | | |
|---|---|---|---|---|
| | **LayerVec** | **GPT-4o** | **Qwen2.5** | **Gemini2.5** |
| *Semantic Fidelity (SF) / Structural Coherence (SC)* | | | | |
| Object Addition | **5.95 / 6.78** | 1.88 / 3.03 | 0.91 / 3.50 | 2.54 / 3.42 |
| Object Removal | **3.72 / 6.34** | 1.99 / 1.95 | 1.22 / 1.46 | 3.67 / 4.21 |
| Bkg. Change | **6.25 / 6.47** | 3.22 / 2.72 | 1.38 / 3.14 | 3.59 / 2.48 |
| Style Transfer | **5.28 / 6.69** | 2.85 / 3.97 | 1.04 / 2.32 | 4.51 / 4.02 |
| Motion Change | **5.13 / 6.23** | 2.15 / 3.32 | 0.85 / 2.94 | 2.25 / 4.85 |
| *Layer-wise CLIP Consistency (LCC)* | | | | |
| Object Addition | **0.2852** | 0.2428 | 0.2329 | 0.2358 |
| Object Removal | **0.2915** | 0.2523 | 0.2311 | 0.2399 |
| Bkg. Change | **0.2922** | 0.2551 | 0.2186 | 0.2376 |
| Style Transfer | **0.2973** | 0.2413 | 0.2108 | 0.2494 |
| Motion Change | **0.3029** | 0.2431 | 0.2177 | 0.2393 |

## 4.4 USER STUDIES

We conducted a user study with 50 design enthusiasts, comparing LayerVec against strong baselines. The results can be seen in App. C. Our method was consistently preferred across three key criteria. For aesthetic quality, over 70% of votes favored our outputs. For both text alignment (in text-to-SVG) and edit instruction alignment (in multimodal-to-SVG), LayerVec was rated as the best or comparable to the best baselines, demonstrating superior semantic consistency and controllability. The study validates that LayerVec produces more visually pleasing, faithful, and practically editable SVGs.

## 4.5 EVALUATION STUDY

In this section, we analyze the core components of our LayerVec framework to validate its design and demonstrate its robustness. We first conduct a detailed ablation study to prove the necessity of each stage in our pipeline, and then we conduct experiments to verify the framework's generalization capabilities across different backbone models.

Table 3: **Ablation and Generalization of the LayerVec framework.** Our framework is applied to two distinct UMMs (OmniGen2 and BAGEL). It consistently and significantly outperforms both naïve synthesis and a strong post-hoc vectorization baseline (VTracer), proving both its efficacy and model-agnostic generalizability.

| Base Model | Framework | SF | SC | LCC |
|---|---|---|---|---|
| | Naïve Prompting | 1.20 | 1.36 | 0.18 |
| OmniGen2 | Synthesis + VTracer | 5.37 | 6.22 | N/A |
| | **+ LayerVec (Ours)** | **5.41** | **6.31** | **0.29** |
| | Naïve Prompting | 1.43 | 2.31 | 0.22 |
| BAGEL-7B | Synthesis + VTracer | 8.14 | 9.03 | N/A |
| | **+ LayerVec (Ours)** | **8.39** | **9.12** | **0.31** |

**Efficacy of the Dual-Stage Pipeline.** To validate our dual-stage design, we first perform an ablation study focusing on our primary backbone, OmniGen2. As detailed in Tab. 3, we compare our full framework against two strong baselines: *(1) Naïve Prompting*, which ablates Stage 1 by directly generating SVG code, and *(2) Synthesis + VTracer*, which ablates Stage 2 by replacing our deconstruction with a non-semantic vectorizer.

The results clearly show that each stage is indispensable. *Naïve Prompting* fails to create meaningful structures, evidenced by a dismal LCC score of 0.18. This confirms that Stage 1's raster guidance is essential for semantic coherence. *Synthesis + VTracer*, while visually faithful (high SC score), is structurally inadequate, yielding an inapplicable LCC (N/A) as it cannot produce semantic layers. Only our complete LayerVec pipeline excels on both visual and structural metrics, proving the necessity of its unique design.

**Generalization Across Backbones.** To demonstrate that the benefits of LayerVec are not specific to a single model architecture, we test its generalization capability. We apply the exact same framework and baselines to a different powerful UMM, BAGEL-7B (Deng et al., 2025).

As shown in Tab. 3, the performance gains are remarkably consistent. On BAGEL-7B, LayerVec again significantly outperforms both *Naïve Prompting* and *Synthesis + VTracer*, especially on the crucial LCC metric (0.31 vs 0.22 and N/A). This consistent superiority across two distinct backbones proves that LayerVec is a robust, model-agnostic framework for generating editable vector graphics.

## 5 APPLICATIONS OF LAYERVEC

Our LayerVec unlocks significant practical applications across various domains of digital content creation by transforming static raster assets into dynamic, structured vector graphics. As demonstrated throughout our experiments, a user can provide any pixel-based image—be it a photograph, a sketch, or a previously generated AI artwork—along with a simple textual instruction, and LayerVec will produce a high-fidelity, semantically layered, and fully editable SVG. This capability streamlines creative workflows in graphic design, iconography, and digital illustration, enabling rapid prototyping, asset recomposition, and non-destructive editing in a manner previously impossible with monolithic raster inputs. We present a diverse gallery of additional application examples in App. B.

## 6 CONCLUSION

We introduce LayerVec, the first framework to successfully generate editable, layered Scalable Vector Graphics from multimodal prompts, bridging a critical gap between automated content creation and professional design workflows. Our key innovation is a dual-stage pipeline that first synthesizes a high-fidelity raster guidance and then employs an MLLM-driven iterative process to deconstruct it into semantically coherent vector layers. To properly evaluate this new paradigm, we contributed a comprehensive benchmark, MUV-Bench, and a novel metric, LCC, which for the first time provides an automated assessment of structural editability of Vector Graphics. Our extensive experiments demonstrate that LayerVec not only significantly outperforms state-of-the-art baselines in producing visually superior and structurally sound SVGs but also exhibits robust generalization across different UMM backbones. We believe this work lays a foundational stone for a new generation of more practical and intuitive vector graphics synthesis tools in practical workflows.

ETHICAL STATEMENT

We have developed our framework in accordance with the ICLR Code of Ethics. Our training data and the newly introduced MUV-Bench are derived from publicly available sources with permissive licenses, and we have made efforts to filter for personally identifiable or offensive content.

REPRODUCIBILITY STATEMENT

The pretrained models of OmniGen2 and BAGEL-7B used in our experiments are publicly available.

To ensure reproducibility, we will also release our code, lora-weights, and the MUV-Bench dataset upon publication. We provide detailed descriptions of our model architectures, tuning procedures, and hyperparameter settings in the Appendix. Additionally, we include scripts for data preprocessing and prompts engineering details.

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

CONTENTS

## A USE OF LLMS

To enhance the quality of the manuscript, Large Language Models (LLMs) were used as an assistive tool for language polishing. The role of the LLM was strictly limited to improving spelling, grammar, and sentence clarity. The final text was reviewed and edited by the authors to ensure it accurately reflects their original ideas and findings.

It is important to note that all ideas, concepts, and the research methodology presented in this paper were exclusively developed and conducted by the authors. LLMs were utilized as an assistive tool, with their role strictly limited to improving the linguistic quality and correcting the grammar of the manuscript. The scientific contributions and findings remain the original work of the authors.

## B MORE RESULTS AND APPLICATIONS OF LAYERVEC

The structured layer organization and multimodal editing capabilities of LayerVec make it particularly well-suited for professional graphic design applications. Unlike traditional raster-based generation methods that produce unorganized outputs, our framework generates semantically meaningful layers that align with standard design practices, enabling seamless integration into existing creative workflows.

For poster design, LayerVec's ability to decompose complex visual compositions into editable components (e.g., text layers, background elements, decorative graphics) allows designers to rapidly iterate on layout arrangements and style variations. The multimodal instruction interface enables intuitive modifications such as "change the background to a warmer tone while keeping the text prominent" or "add more visual emphasis to the call-to-action elements."

Fig. 7 illustrates how LayerVec can generate posters with distinct visual themes while maintaining a coherent structure that allows for easy adjustments. The framework's semantic layer extraction ensures that each component can be independently manipulated, facilitating rapid design iterations and refinements.

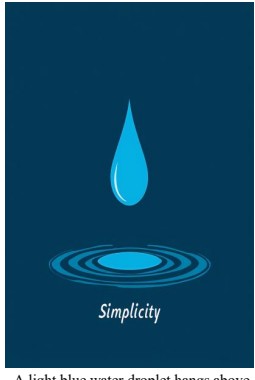 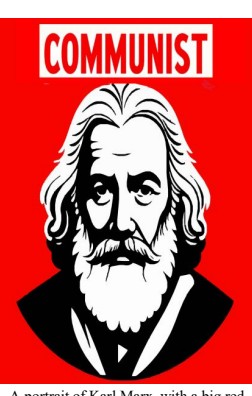 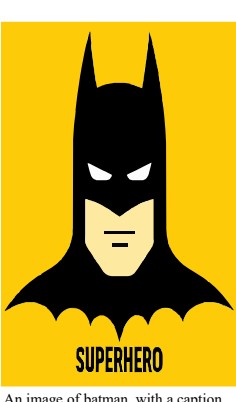 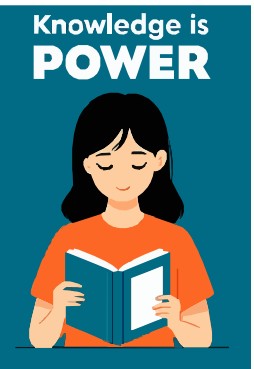

A light blue water droplet hangs above stylized, ripples on a water surface, with caption "Simplicity" below

A portrait of Karl Marx, with a big red title "Communist" on the top

An image of batman, with a caption "SUPERHERO" under him.

A poster of a girl reading a book with a big caption "Knowledge is power" on the top.

Figure 7: SVG posters generated by LayerVec.

Similarly, in logo creation, the framework's semantic layer extraction ensures that individual design elements (symbols, text, decorative elements) remain independently manipulable, facilitating brand identity variations and responsive design adaptations. More results of SVG icons generated from LayerVec can be seen in Fig. 8.

In creative workflows, multimodal inputs can significantly enhance the quality of the generated outputs. By leveraging both visual and textual information, designers can provide richer context and more precise instructions, leading to results that better align with their creative vision. This capability is particularly valuable in iterative design processes, where quick adjustments and refinements are often necessary. The ability to seamlessly integrate multimodal inputs into the design workflow

not only streamlines the creative process but also empowers designers to explore a wider range of possibilities and achieve higher-quality outcomes(See Fig. 9).

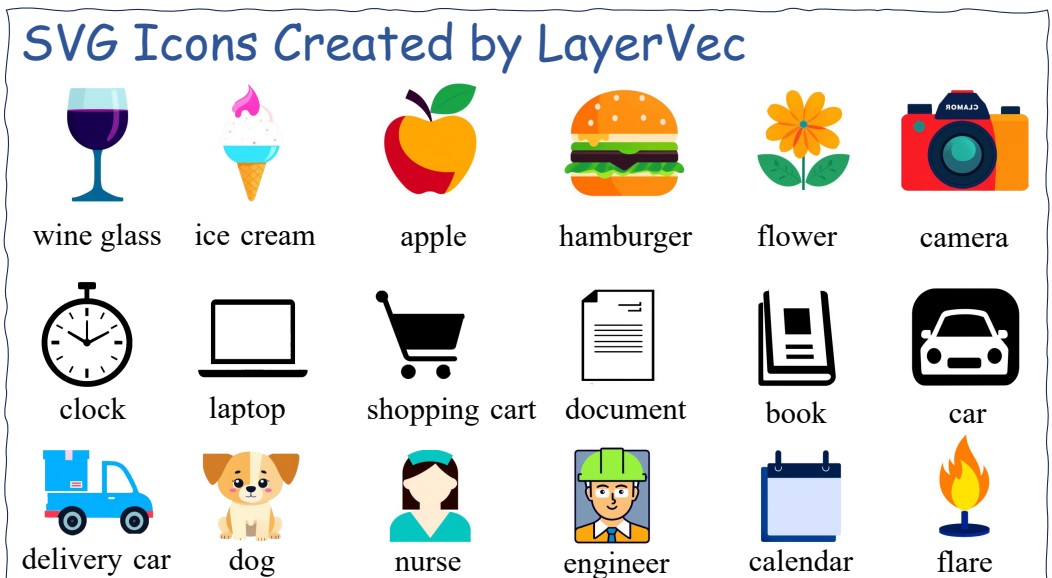

Figure 8: SVG Icons generated by LayerVec.

## C  USER STUDY

Fig. 10 shows the result of user study. We conduct a user study across 10 designers with expertise in graphic design. The study aims to evaluate the usability and effectiveness of LayerVec in real-world design scenarios.

## D  DETAILS OF OMNIGEN2

OmniGen2 is a versatile and open-source multimodal generative model designed to provide a unified solution for a diverse range of generation tasks, including text-to-image synthesis, image editing, and in-context generation. It builds upon its predecessor, OmniGen (Xiao et al., 2025), by introducing significant architectural and data-centric innovations to enhance performance and capability. This section provides a detailed overview of its core components, training methodology, and novel mechanisms, as described in the original paper.

### D.1  MODEL ARCHITECTURE

A key design principle of OmniGen2 is the decoupling of its text and image generation pathways to preserve the strong, pre-existing capabilities of its foundational Multimodal Large Language Model (MLLM). Unlike fully unified architectures where parameters are shared, OmniGen2 employs two distinct transformer-based modules with unshared parameters.

- **Multimodal Large Language Model (MLLM):** The core of OmniGen2's understanding and text generation is a frozen, pre-trained MLLM, specifically initialized from Qwen2.5-VL-3B. This module processes interleaved sequences of text and image inputs. For image inputs, it utilizes a Vision Transformer (ViT) to encode visual information into high-level semantic embeddings. The MLLM is responsible for interpreting user instructions and generating textual responses autoregressively.
- **Diffusion Transformer:** Image generation is handled by a separate diffusion transformer, which is trained from scratch and comprises approximately 4 billion parameters. This

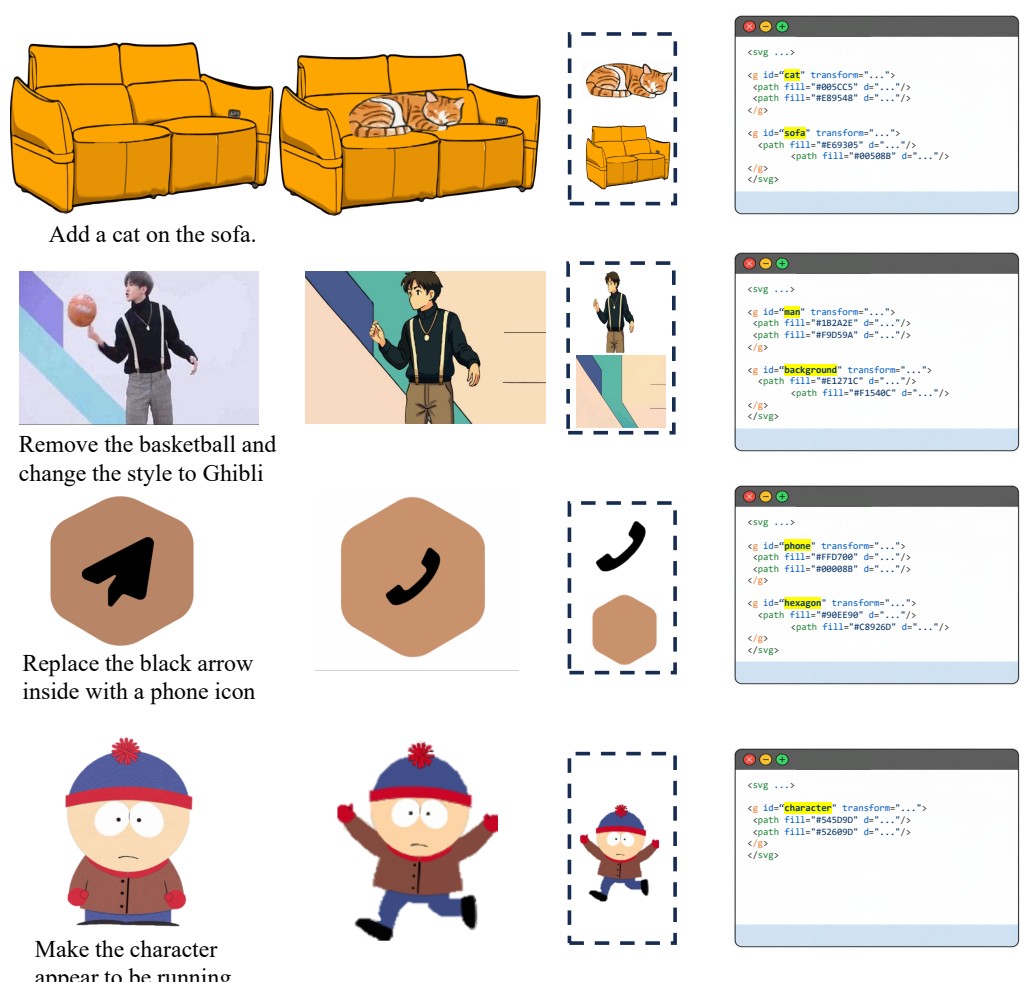

Figure 9: multimodal-to-SVG results generated by LayerVec.

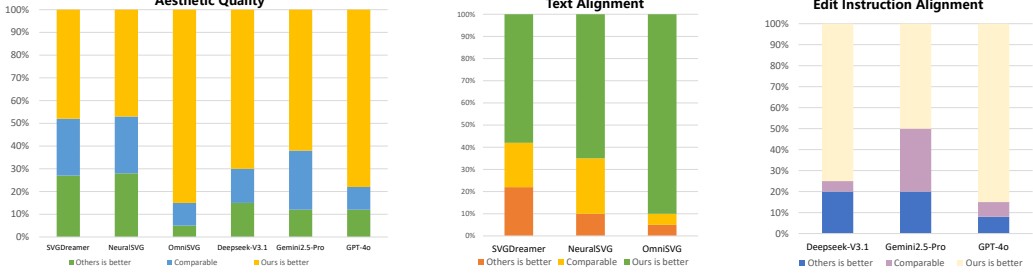

Figure 10: User studies.

module is conditioned on the hidden states produced by the MLLM in response to a textual prompt. To explicitly trigger image synthesis, a special token, `"<|img|>"`, is introduced into the vocabulary. When the MLLM generates this token, the diffusion process is initiated.

- **Dual Image Encoders:** OmniGen2 employs a dual-encoder strategy to balance semantic understanding with high-fidelity generation. While the ViT encoder provides semantic context to the MLLM, a Variational Autoencoder (VAE) is used to encode input images into a latent space suitable for the diffusion transformer. Crucially, the VAE-derived features are

fed directly into the diffusion model and not the MLLM. This design choice prevents the low-level VAE features from degrading the MLLM's inherent understanding capabilities and avoids the architectural complexity of dual-encoding within the MLLM itself.

## D.2 KEY INNOVATIONS

**Omni-RoPE Positional Embedding.** To effectively handle complex tasks like multi-image editing and in-context learning, OmniGen2 introduces a novel multimodal rotary position embedding named **Omni-RoPE**. This 3D positional encoding decomposes position information into three components:

1. **Sequence and Modality Identifier ($id_{seq}$):** This component distinguishes between different modalities and sequences. For text tokens, it functions as a standard 1D positional index. For image tokens, all tokens belonging to the same image share a constant, unique identifier, treating the entire image as a single semantic unit.

2. **2D Spatial Height Coordinate ($h$):** The normalized vertical position for each image token.

3. **2D Spatial Width Coordinate ($w$):** The normalized horizontal position for each image token.

This structure allows the model to unambiguously differentiate between multiple input and output images via their unique $id_{seq}$, while the locally computed spatial coordinates $(h, w)$ ensure consistency and accurate region preservation during editing tasks.

## D.3 DATA CONSTRUCTION AND TRAINING STRATEGY

Recognizing the limitations of existing open-source datasets, the authors of OmniGen2 developed comprehensive data construction pipelines to generate high-quality training data, particularly for advanced editing and in-context tasks.

- **In-Context Data from Videos:** To create data for subject-driven generation and editing, video frames are leveraged. The pipeline identifies a primary subject in a base frame using an MLLM, segments and tracks it across subsequent frames using GroundingDINO and SAM2, and then uses inpainting/outpainting models (e.g., FLUX.1-Fill-dev) to generate novel backgrounds or create editing pairs (e.g., transplanting a subject from a context image into a target image).

- **High-Quality Image Editing Data:** To overcome instruction-image misalignment in existing datasets, OmniGen2's data pipeline starts with high-quality images, applies a powerful inpainting model to randomly fill a masked region, and *then* uses a strong MLLM (Qwen2.5-VL) to generate a precise editing instruction that describes the transformation between the original and inpainted images. This ensures high fidelity and accurate instruction-following supervision.

The training process is staged. The MLLM's parameters are largely kept frozen to preserve its understanding capabilities. The diffusion model is first trained from scratch on text-to-image generation and subsequently trained on a mixed-task objective including editing and in-context data.

## D.4 REFLECTION MECHANISM

OmniGen2 introduces a reflection mechanism to incorporate reasoning and self-correction into the image generation process. This is facilitated by a curated reflection dataset.

1. The model first generates an image based on an initial user instruction.

2. A powerful external MLLM (e.g., Doubao-1.5-pro) then assesses the generated image against the instruction.

3. If deficiencies are found (e.g., incorrect object count, color, or composition), the MLLM generates a textual "reflection" that identifies the error and proposes a specific modification.

4. This sequence—(instruction, initial image, reflection text, corrected image)—forms a multi-turn training sample.

By fine-tuning on this data, OmniGen2 learns to generate reflective text and iteratively refine its image outputs, improving its instruction-following and reasoning abilities. All model parameters are unfrozen during this phase of training.

## E  ALGORITHM FOR ITERATIVE SCENE DECONSTRUCTION

This section provides a detailed, step-by-step walkthrough of our Iterative Scene Deconstruction workflow, as described in Section Sec. 3.3 of the main paper. We use the prompt "an astronaut riding a horse on the moon" as a running example to illustrate the process.

**Step 1: Scene Planning and Z-Order Inference.**  The process begins with the MLLM, $\mathcal{M}_{\text{plan}}$, analyzing the user prompt $P_T$ and the initial synthesized raster image $I_0$. It is tasked with both identifying the key semantic entities and inferring their occlusion hierarchy. The MLLM employs chain-of-thought reasoning to output an ordered list of entities, $O$, representing the scene's structure from foreground to background.

The resulting plan, $O = ($"astronaut", "horse", "moon"$)$, now dictates the sequence for the deconstruction loop. The initial image state is set to $I_{i-1} \leftarrow I_0$.

**Step 2: Iteration 1 - Extracting the "astronaut" Layer.**  The loop begins with the top-most entity, $o_1 = $ "astronaut". First, the MLLM, acting as a visual grounder $\mathcal{M}_{\text{detect}}$, localizes the astronaut in the current image $I_0$ to produce a bounding box $B_1$. This box is then used to prompt a segmentation model $\mathcal{S}$ to yield a precise mask $m_1$. The raster layer $\hat{l}_1$ is then extracted via element-wise multiplication: $\hat{l}_1 = I_0 \odot m_1$.

**Step 3: Iteration 1 - Inpainting for Occlusion.**  To prepare for the next iteration, the background occluded by the astronaut must be synthesized. The MLLM, now in an "art director" role, generates a context-aware inpainting prompt $P_{\text{inp}}^{(1)}$ based on the remaining entities in the plan, $O[2 :] = ($"horse", "moon"$)$. Our controllable synthesis engine $\mathcal{D}_{\text{inp}}$ is then invoked to fill the region defined by mask $m_1$.

The output of this step, $I_1$, is a new, high-fidelity image depicting a complete horse on the moon, with no trace of the astronaut. This image now serves as the input for the next iteration, $I_{i-1} \leftarrow I_1$.

**Step 4: Subsequent Iterations and Final Composition.**  The process repeats for the next entity, $o_2 = $ "horse". The layer $\hat{l}_2$ is extracted from the image $I_1$ (which already has a complete horse), and the background is then inpainted to remove the horse, leaving only the complete moon surface as the final image state, $I_2$. This final state becomes the background layer, $\hat{l}_{\text{bg}} = I_2$. The final set of deconstructed raster layers, $\hat{\mathcal{L}} = \{\hat{l}_1, \hat{l}_2, \hat{l}_{\text{bg}}\}$, now contains a complete, non-overlapping representation of the entire scene. Each of these layers is then independently vectorized and composed to form the final, editable SVG document.

In summary, 1 outlines the complete algorithm for Iterative Scene Deconstruction, detailing how each step builds upon the previous one to achieve a structured, layered representation of complex visual scenes.

---

**Algorithm 1** Iterative Deconstruction

---

1: **Input:** Image $I_0$, Prompt $P_T$
2: $O \leftarrow \mathcal{M}_{\text{plan}}(P_T, I_0)$
3: $I_{i-1} \leftarrow I_0$ ; $\hat{\mathcal{L}} \leftarrow \emptyset$
4: **for** $i = 1$ to $|O|$ **do**
5:     $o_i \leftarrow O[i]$
6:     $m_i \leftarrow (\mathcal{S} \circ \mathcal{M}_{\text{detect}})(I_{i-1}, o_i)$
7:     $\hat{l}_i \leftarrow I_{i-1} \odot m_i$
8:     $\hat{\mathcal{L}} \leftarrow \hat{\mathcal{L}} \cup \{\hat{l}_i\}$
9:     **if** $i < |O|$ **then**
10:         $P_{\text{inp}}^{(i)} \leftarrow \mathcal{M}_{\text{plan}}(O[i+1:])$
11:         $I_i \leftarrow \mathcal{D}_{\text{inp}}(I_{i-1}, m_i, P_{\text{inp}}^{(i)})$
12:         $I_{i-1} \leftarrow I_i$
13:     **end if**
14: **end for**
15: $\hat{l}_{\text{bg}} \leftarrow I_{|O|}$
16: **return** $\hat{\mathcal{L}} \cup \{\hat{l}_{\text{bg}}\}$

---

# F EXPERIMENTAL DETAILS

## F.1 LoRA PARAMETERS

To specialize our foundational model for the vector graphics domain, we employed a lightweight LoRA fine-tuning strategy. The experiment was conducted on a single NVIDIA L40 GPU with 40GB of memory.

## F.2 TRAINING CONFIGURATION

The fine-tuning process was conducted with a focus on efficiency and stability. We used a global batch size of 1, with a learning rate of $1 \times 10^{-4}$ and the `timm_constant_with_warmup` scheduler. Gradient accumulation and checkpointing were enabled to manage memory usage. The training was performed using `bf16` mixed precision for a total of 8,000 steps. Key training parameters are summarized in Tab. 4.

Table 4: Key training and optimizer hyperparameters for LoRA fine-tuning.

| Parameter | Value |
|---|---|
| ***General Training*** | |
| Max Training Steps | 8000 |
| Global Batch Size | 1 |
| Gradient Accumulation Steps | 1 |
| Mixed Precision | `bf16` |
| Gradient Checkpointing | Enabled |
| ***Optimizer (Adam)*** | |
| Learning Rate (`lr`) | $1 \times 10^{-4}$ |
| LR Scheduler | `timm_constant_with_warmup` |
| Warmup Steps | 500 |
| Adam $\beta_1$ / $\beta_2$ | 0.9 / 0.95 |
| Weight Decay | 0.01 |
| Max Grad Norm | 1.0 |
| ***LoRA Specific*** | |
| LoRA Rank (`lora_rank`) | 8 |
| LoRA Dropout | 0.0 |

### F.3 MODEL AND DATA CONFIGURATION

Our framework is built upon pre-trained components. The VAE was initialized from `FLUX.1-dev`, and the text encoder from `Qwen2.5-VL-3B-Instruct`. The main model architecture was based on our pre-trained `OmniGen2`. Tab. 5 details the core architectural hyperparameters of the Diffusion Transformer that was fine-tuned. For data handling, input images were processed with a maximum resolution of $1024 \times 1024$ pixels, and a dropout probability of 0.5 was applied to reference images to improve robustness.

Table 5: Model architecture and data processing hyperparameters.

| Parameter | Value |
|---|---|
| ***Pre-trained Components*** | |
| VAE Model | `FLUX.1-dev` |
| Text Encoder | `Qwen2.5-VL-3B-Instruct` |
| Base Model | `OmniGen2` |
| ***Diffusion Transformer Architecture*** | |
| Hidden Size | 2520 |
| Number of Layers | 32 |
| Attention Heads | 21 |
| Patch Size | 2 |
| Input Channels | 16 |
| ***Data Processing*** | |
| Maximum Text Tokens | 888 |
| Max Output Pixels | $1024 \times 1024$ |
| Reference Image Dropout | 0.5 |

## G DETAILS OF VTRACER

VTracer is a raster-to-vector conversion algorithm developed by Vision Cortex. Its pipeline consists of three major stages: path generation, path simplification, and curve fitting. This design balances high fidelity with compact vector representations, making it suitable for large-scale or high-resolution image processing. In this section we briefly summarize the algorithmic details.

### PATH GENERATION

The raster image is first decomposed into clusters of pixels with identical labels. For each cluster, a boundary *walker* traverses the edges of pixels to form polygonal paths. Consecutive steps in the same direction are merged to reduce redundancy.

### PATH SIMPLIFICATION

Two strategies are employed to reduce the staircase-like artifacts inherent in raster boundaries:

- **Staircase Removal**: Each middle point on a three-point segment is evaluated by the signed area of the corresponding triangle. If the area indicates collinearity, the middle point is removed.
- **Error-Penalized Simplification**: For a subpath, its approximation by a straight line is accepted only if the accumulated distance penalty of intermediate points is below a threshold.

### CURVE FITTING AND SMOOTHING

The simplified polygon is then smoothed by a modified 4-point subdivision scheme. Special handling is applied at *corner points* (detected by angle thresholds) to preserve sharp features. The subdivided path is segmented at splice points, determined either by inflection detection or accumulated angular deviation, before Bezier curves are fitted to each segment.

