# OpenReview forum: "Toward Editable Vector Graphics: Layered SVG Synthesis from Multimodal Prompts"
_ICLR.cc/2026/Conference — ICLR 2026 Conference Withdrawn Submission_

### Official Review · Reviewer_NzvV · 2025-10-27

**Soundness:** 2
**Presentation:** 3
**Contribution:** 2
**Rating:** 2
**Confidence:** 4

**Summary:**

The paper introduces LayerVec, a framework for generating editable, layered SVGs from multimodal prompts (text + image). It employs a dual-stage pipeline—first synthesizing a raster guidance image, then iteratively deconstructing it into semantically coherent vector layers—enabling clean, structured, and editable SVG outputs. The authors also propose a new benchmark (MUV-Bench) and metric (Layer-wise CLIP Consistency) to evaluate structural editability, demonstrating strong performance and model-agnostic generalization across unified multimodal models. However, the claim of being the first to explore SVG generation from multimodal inputs is not correct, there are existing works on this research line. Please see my detailed comments below in the weaknesses section.

**Strengths:**

(1) The paper proposes a SVG generation framework from multimodal inputs together with some evaluation tools such as MUV-Bench dataset and the LCC metric. However, many of these claimed new proposals are already existing.

**Weaknesses:**

(1) The claim of being the first one to explore SVG generation from multimodal inputs is not right - there exist works that fall in this research line - that is they consider multimodal inputs and generate SVGs [A,B,C]. I understand this particular paper is motivated by the layered design of an SVG, however, in my understanding SVG is layered by its original design. So I don't understand the novelty in the proposed approach. Also, these works [A,B] are quite related to the proposed ones, so should have been discussed in the related works.

(2) The paper needs more details about the proposed benchmark. At the moment, we only know that the dataset comprises 500 carefully curated tasks created from 50 source images and 10 editing instructions each. However, examples of such images and editing instructions are not there in the paper including the supplementary material - this should be in the form of qualitative examples.

(3) The proposed evaluation metric LCC is a combination of CLIP scores across different layers in an SVG - this is a trivial measurement. I am unsure if this should be considered as an impactful metric for this body of works.

[A] Zhang et al., Text-Guided Vector Graphics Customization, SIGGRAPH Asia, 2023.

[B] StarVector: Generating Scalable Vector Graphics Code from Images and Text, CVPR, 2025.

[C] Yang et al., OmniSVG: A Unified Scalable Vector Graphics Generation Model, NeurIPS, 2025.

**Questions:**

(1) How computationally demanding is the iterative scene deconstruction stage, and what is its average runtime compared to single-pass SVG generation methods? Could the authors discuss whether LayerVec can be optimized for real-time or interactive design use cases?

(2) The framework currently employs VTracer as a post-processing step. Have the authors considered integrating a learnable or differentiable vectorization module to make the system fully end-to-end? Does the use of VTracer introduce any artifacts or inconsistencies in the layer boundaries that affect downstream editability?

---

### Official Review · Reviewer_3QJ2 · 2025-10-31

**Soundness:** 4
**Presentation:** 4
**Contribution:** 3
**Rating:** 8
**Confidence:** 5

**Summary:**

This work presents LayerVec, a novel framework for synthesizing editable, layered SVGs from multimodal prompts. The core idea is a dual-stage pipeline: a Unified Multimodal Model first generates a raster guidance image, which is then fed into an innovative iterative deconstruction process. This MLLM-guided process peels the image into semantic raster layers, which are then vectorized independently.

**Strengths:**

1. The paper tackles a highly relevant and underserved problem. The goal of producing editable, layered vector graphics, especially from multimodal inputs (e.g., "add a hat to this image"), directly aligns with real-world creative workflows in a way that existing text-to-"flat-SVG" models do not.
2. The 2-stage pipeline is very clever. It sidesteps the difficulty of direct, layered vector synthesis from a prompt. The iterative deconstruction process is a robust and well-reasoned approach to achieving semantic separation.
3. The method is well-backed by extensive experimentation and benchmarking.

**Weaknesses:**

1. The framework is a multi-step cascade and is prone to error-propagation. A failure at any single step could lead to a catastrophic failure in the final output.
2. This pipeline appears to be computationally expensive and slow. It requires multiple calls to large models (a DiT, an MLLM, etc) per layer. This iterative nature could make it prohibitively slow for complex scenes, which is a significant practical limitation not addressed in the paper.

**Questions:**

1. Could you please elaborate on the inference time and computational cost? How does the framework's runtime scale with the number of detected semantic layers in a scene?
2. How does the pipeline handle failures or low-confidence outputs at intermediate steps?
3. The LCC metric is excellent. Does it, however, capture the internal path quality of a vectorized layer?

---

### Official Review · Reviewer_rYD6 · 2025-10-31

**Soundness:** 2
**Presentation:** 3
**Contribution:** 2
**Rating:** 4
**Confidence:** 3

**Summary:**

This paper tackles the problem of generating editable, layered SVG graphics from multimodal prompts. The authors propose LayerVec, a two-stage pipeline built on top of large unified multimodal models (UMMs). In Stage 1, a fine-tuned UMM (a diffusion transformer guided by an LLM) produces a high-fidelity raster guidance image from the combined text+image prompt. In Stage 2, an iterative scene deconstruction process uses the LLM as a planner: it identifies foreground entities in front-to-back order and, for each entity, proposes a bounding box, applies a segmentation model to extract that object’s mask, and then inpaints the occluded background before proceeding to the next object.

**Strengths:**

1. **Addresses an important gap:** The paper correctly notes that existing SVG generation methods handle only single-modality inputs and ignore layered structure. In real design workflows, multimodal inputs are common, and layered SVGs are needed for object-level editing. LayerVec ambitiously targets this underexplored scenario, which is a meaningful contribution in graphics.

2. **Innovative two-stage design:** The combination of a generative prior and an LLM-driven decomposition is clever. By first unifying the prompt into a concrete raster image, the method leverages powerful vision-and-language models to resolve ambiguities. Then, the iterative LLM-guided segmentation and inpainting effectively layered objects one by one. This modular design separates appearance from structure and explicitly produces semantic layers, which is novel compared to prior work that often yields a flat SVG.

**Weaknesses:**

1. Although the authors tested LayerVec on two UMMs, both (OmniGen2 and BAGEL-7B) share a similar high-level design (an LLM plus a diffusion generator). The paper concludes that LayerVec proves model-agnosticism, but it remains unclear how it would transfer to substantially different models (e.g., a single-stream vision-language model, a GAN-based image generator, or a purely code-generating LLM).

2. The gains reported are also modest: e.g., on OmniGen2, the Semantic Fidelity score only rises from 5.37 to 5.41 after adding the layering stage, suggesting limited benefit beyond enabling layers. The claim of broad generality seems weakly supported by only two similar backbones.

3. LayerVec’s key novelty is the iterative decomposition into layers. However, the paper provides little quantitative analysis of how accurately this segmentation works. It relies on an unspecified segmentation model S (given an LLM-proposed box). If the box is imprecise or the object has complex textures, the mask might be wrong. Errors could accumulate over iterations (incorrect inpainting of occluded regions). The paper does not report any failure cases or metrics for segmentation accuracy.

4. We lack evidence that the layers correspond cleanly to semantic objects beyond qualitative impressions. The iterative process could also be time-consuming, but runtime or efficiency is not discussed. In short, the quality and robustness of the layer extraction step are assumed rather than proven.

5. While LCC is presented as essential for assessing structural editability and is LayerVec’s key advantage, it is explicitly noted that major categories of state-of-the-art baselines cannot be evaluated using this metric.

**Questions:**

1. Have you evaluated how often the correct objects are identified and segmented?

2. What happens if the LLM’s plan misses an object or proposes an incorrect box?

3. How sensitive is the method to the choice of backbone?

4. Do you have any empirical evidence (e.g. correlation) that high LCC indeed predicts better manual editability? Could there be cases where LCC is high but editing fails (or vice versa)?

---

### Official Review · Reviewer_J2k8 · 2025-11-02

**Soundness:** 2
**Presentation:** 2
**Contribution:** 1
**Rating:** 2
**Confidence:** 3

**Summary:**

This paper introduces LayerVec, a framework for generating layered SVG from multimodal prompts. ​The approach uses an iterative procedure to decompose a generated raster image into semantic layers, leveraging a chain-of-thought reasoning process. ​The framework employs a UMM to analyze the multimodal prompt, plan the scene, and guide the extraction of semantic entities layer by layer.  Each layer is segmented, inpainted, vectorized using special models.

**Strengths:**

+ The first attempt to generate SVG layers using UMM
+ Created a MUV-bench consisting of 50 raster images, each paired with 10 editing instructions
+ Proposed a new layer-wise CLIP consistency evaluation metric

**Weaknesses:**

- Biggest shortcoming is this paper is not much related to SVG vector generation. Most of the system is about image generation, segmentation and inpainting. Only the last step converts layers to SVG using off-the-shelf tool. The curated benchmark is also mostly on image tasks, not so related to editable SVG layer generation as the paper promises.
- The proposed benchmark is very small, compared to many existing image datasets.
- It is not clear why the DiT part of UMM is tuned, not the other components. Some motivation should be given on what and why certain components need to be enhanced from a generalist UMM.

**Questions:**

- why DiT is fine-tuned, while other parts are not?
- is it possible to do all imaging tasks (generation/segmentation/inpainting) with the same UMM? what makes this challenging
- any special considerations for SVG generation versus photo layer generation?

---

### Note · Authors · 2025-11-12

I have read and agree with the venue's withdrawal policy on behalf of myself and my co-authors.